# MAVIN: Multi-Action Video Generation with Diffusion Models via Transition Video Infilling

## Abstract

Diffusion-based video generation has achieved significant progress, yet generating multiple actions that occur sequentially remains a formidable task. Directly generating a video with sequential actions can be extremely challenging due to the scarcity of fine-grained action annotations and the difficulty in establishing temporal semantic correspondences and maintaining long-term consistency. To tackle this, we propose an intuitive and straightforward solution: splicing multiple single-action video segments sequentially. The core challenge lies in generating smooth and natural transitions between these segments given the inherent complexity and variability of action transitions. We introduce MAVIN (Multi-Action Video INfilling model), designed to generate transition videos that seamlessly connect two given videos, forming a cohesive integrated sequence. MAVIN incorporates several innovative techniques to address challenges in the transition video infilling task. Firstly, a consecutive noising strategy coupled with variable-length sampling is employed to handle large infilling gaps and varied generation lengths. Secondly, boundary frame guidance (BFG) is proposed to address the lack of semantic guidance during transition generation. Lastly, a Gaussian filter mixer (GFM) dynamically manages noise initialization during inference, mitigating train-test discrepancy while preserving generation flexibility. Additionally, we introduce a new metric, CLIP-RS (CLIP Relative Smoothness), to evaluate temporal coherence and smoothness, complementing traditional quality-based metrics. Experimental results on horse and tiger scenarios demonstrate MAVIN's superior performance in generating smooth and coherent video transitions compared to existing methods.

## 1 Introduction

The evolution of video generation models has been significantly shaped by the advent of diffusion-based techniques, offering unprecedented fidelity and temporal coherence in video synthesis (Blattmann et al., 2023a; Ho et al., 2022a; Singer et al., 2022; Wang et al., 2023b; Ho et al., 2022b). However, these models often struggle to generate videos that encompass multiple actions or adhere to complex instructions, and typically produce relatively short clips, limiting their use in scenarios requiring longer, multi-action sequences.

Generating multi-action videos directly presents numerous unresolved challenges. Firstly, the lack of fine-grained action-level annotations in existing large-scale video datasets hampers model training. Secondly, multi-action sequences, involving extended durations and significant motion ranges, challenge models to maintain spatiotemporal consistency throughout the video. The structural characteristics of video U-Net models further complicate complex temporal semantic correspondence modeling. To circumvent these challenges, in this paper, we propose an innovative approach for generating multi-action videos by integrating several single-action video clips. This process entails two fundamental steps: first, the production of various video clips featuring the same subject engaging in distinct actions; second, the concatenation of these clips through action transitions. While the first step has been facilitated by recent advancements in text-conditioned image-to-video (TI2V) generation (Dai et al., 2023; Girdhar et al., 2023; Xing et al., 2023; Ren et al., 2024; Zhang et al., 2024; Wei et al., 2023), the second step remains understudied. To this end, we introduce MAVIN

(**M**ulti-**A**ction **V**ideo **IN**filling model), a transition model designed to infill an intermediate video clip between two adjoining clips, ensuring a fluid and seamless transition.

This task requires meticulous attention to overall motion consistency and smoothness. Therefore, MAVIN is trained with consistent conditioning on the reference videos. To manage the potential for substantial motion gaps and the requirement for flexible infilling lengths, we utilize a variable-length sampling strategy. The performance of MAVIN is further enhanced by boundary frame guidance (BFG) and a Gaussian filter mixer (GFM). BFG leverages high-level semantic features from the boundary frames of input videos to guide the video infilling process, ensuring visual coherence throughout the transition. Meanwhile, GFM dynamically manages the introduction and modulation of noise during inference, improving generation fidelity while maintaining flexibility. Our method is trained in a self-supervised manner, eliminating the need for finely annotated video-text transcriptions.

Moreover, existing metrics for evaluating video generation primarily focus on visual quality and often overlook temporal coherence, which is crucial for assessing action transitions. To address this gap, we introduce a new metric, CLIP-RS (CLIP Relative Smoothness), specifically designed to evaluate the temporal consistency and smoothness of transition videos. This metric complements traditional quality-based metrics and provides a comprehensive evaluation of our model's performance. Experimental results conducted on two distinct animal scenarios—horses and tigers—demonstrate our method's superior performance in generating smooth and natural video transitions over existing methods, both in qualitative and quantitative assessments.

## 2 RELATED WORK

**Text-to-Video Generation.** Text-to-Video (T2V) studies have shifted their focus from GAN-based models (Fox et al., 2021; Brooks et al., 2022; Tian et al.; Shen et al., 2023) and auto-regressive models (Ge et al., 2022; Hong et al., 2022; Le Moing et al., 2021; Yan et al., 2021) to diffusion models (Zhang et al., 2023; Jeong et al., 2023; Singer et al., 2022; Ge et al., 2023; He et al., 2022; Zhou et al., 2022; Yang et al., 2023; Ho et al., 2022a), attributed to their superiority in generation quality, training stability, and condition flexibility. Foundation T2V models such as ModelScopeT2V (Wang et al., 2023a) and VideoCrafter (Chen et al., 2023a; 2024) are trained on large-scale captioned datasets, possessing rich motion priors and text-motion correspondences. Nevertheless, challenges persist in generating actions that fully adhere to complex text descriptions.

**Image-to-Video Generation.** Generating videos solely from text prompts leads to a high degree of randomness in the appearance of each generation, thereby limiting its range of applications. Image-to-Video (I2V) generations, on the other hand, animate a user-input image by leveraging the motion priors learned from video-only datasets (Blattmann et al., 2023a; Guo et al., 2023; Jin et al., 2024; Wu et al., 2023b) and have demonstrated the ability to generate high-fidelity and aesthetically pleasing videos. However, they often exhibit limitations in the form of minor and uncontrollable motion patterns. Considering these issues, many studies have begun to focus on text-conditioned image-to-video generation (TI2V) synthesis, which involves generating videos from a reference image, coupled with a text prompt indicating how the image should be animated. Videos generated in this manner typically use the provided image as the initial frame (Girdhar et al., 2023; Dai et al., 2023; Zeng et al., 2023; Ren et al., 2024; Gong et al., 2024) or retain its appearance identity and characteristics (Wei et al., 2023; Zhang et al., 2024; Xing et al., 2023), while performing the motion described in the text. There has also been a stream of works that further specialize in motion controllability by integrating extra controlling signals (Chen et al., 2023b; Kandala et al., 2024; Shi et al., 2024; Ma et al., 2024).

**Generative Video Interpolation.** Diffusion models have also gained momentum in video interpolation, challenging traditional methods that rely on optical flow computation and frame blending techniques. MCVD (Voleti et al., 2022) and RaMViD (Höppe et al., 2022) adopt diffusion-based models with random frame masking, making it capable of handling a range of video generative modeling tasks, including video prediction and interpolation. LDMVFI (Danier et al., 2024) claims to be the first effort solving video interpolation using latent diffusion models and has achieved superior perceptual quality compared to traditional models. However, these works are primarily centered on standard video frame interpolation tasks, where the motions are less ambiguous and straightforward. A concurrent work (Jain et al., 2024) delves into large and challenging motions by interpolating 7 frames with an approximate stride of 3. It utilizes a cascaded framework where the interpolation occurs at a low-resolution pixel level and is subsequently upsampled with a super-resolution model.

However, the absence of open-source availability of the model and data renders further evaluation under our task scenario unfeasible. SEINE (Chen et al., 2023c) explores diffusion-based scene transition, where the model can generate a smooth transition from the start image depicting one scene to the end image representing another.

# 3 METHODOLOGY

## 3.1 PRELIMINARIES

**Latent diffusion model** (LDM) (Rombach et al., 2022) is a diffusion model (DM) (Ho et al., 2020) variant that operates on the compressed latent space instead of the pixel space, and has exhibited its strong efficacy in image generation. LDM first encodes an input image sample $x_0$ into a clean latent code $z_0 = \mathcal{E}(x_0)$ using a VAE (Kingma & Welling, 2013; Esser et al., 2021) encoder $\mathcal{E}(\cdot)$. The latent code then undergoes a forward diffusion process, where it is incrementally perturbed with Gaussian noise following a Markov chain

$$q(z_t|z_{t-1}) = \mathcal{N}(z_t; \sqrt{1-\beta_t}z_{t-1}, \beta_t I), \tag{1}$$

where $t \in \{1, \ldots, T\}$, and $T$ is the number of total forward diffusion steps. $\beta_t$ controls the noise strength at each step. By rewriting $\bar{\alpha}_t := \prod_{i=1}^{t}(1 - \beta_i)$, this formula can be simplified as

$$z_t = \sqrt{\bar{\alpha}_t}z_0 + \sqrt{1-\bar{\alpha}_t}\epsilon, \quad \epsilon \sim \mathcal{N}(0, I). \tag{2}$$

A U-Net (Ronneberger et al., 2015) model parameterized with $\theta$ works as a noise prediction function $\epsilon_\theta(\cdot)$ to predict the added noise $\epsilon$ given the time step $t$ and condition $c$ (e.g. text prompt). The training objective can be formulated as

$$\arg\min_\theta \mathbb{E}_{z_0,\epsilon,t,c}\|\epsilon - \epsilon_\theta(z_t, t, c)\|_2^2. \tag{3}$$

**Video latent diffusion model** (VLDM) (Blattmann et al., 2023b; Ho et al., 2022b; Esser et al., 2023; Wang et al., 2024) inflates the U-Net model into a 3D architecture by inserting temporal modules, making it capable of handling video data. Given an encoded video latent representation $z \in \mathbb{R}^{n \times h \times w \times c}$ where $n$ is the number of frames in the video; $h$ and $w$ denote the height and width of the latent code; and $c$ is the dimension of the latent space, the model performs spatial operations over the $h \times w$ space and temporal operations along the $n$ axis. The spatiotemporal structure empowers the model to manage spatial and temporal dependencies in a coordinated manner, facilitating the generation of coherent and high-quality video sequences.

## 3.2 PROBLEM FORMULATION AND CHALLENGES

**Problem formulation.** The proposed transition video infilling task is a specialized form of video interpolation that deals with long ranges and large motions, with the input being two videos. The objective of this task is to generate a transition video given two videos, one preceding and one following, thereby seamlessly connecting the two. Given an encoded preceding video latent $z_0^{\mathcal{P}} = \{z_0^0, \ldots, z_0^s\}$ and a following video latent $z_0^{\mathcal{F}} = \{z_0^e, \ldots, z_0^{L-1}\}$, the model aims to generate an intermediate latent $z_0^{\mathcal{I}} = \{z_0^{s+1}, \ldots, z_0^{e-1}\}$, where $s$ is the end frame index of the preceding video and $e$ is the start frame index of the following video. We term these two frames as the *boundary frames* for simplicity and clarity. $L$ is the length of the integrated video after infilling.

**Challenges and remedies.** The novel nature of this task presents new challenges. In this section, we briefly outline the challenges and our solutions, with a detailed elaboration to follow in the next section.

The first challenge lies in temporal dependency modeling, which should support generating a transition video with potentially large motion gaps while maintaining motion consistency. Existing works (Chen et al., 2023c; Höppe et al., 2022) typically adopt a BERT-like masking strategy for conditional modeling. However, such approaches are not effective for learning long-span motion patterns as prediction targets and references appear alternately on the temporal axis. To address this issue, we propose to consistently apply noise to *a consecutive subsequence* of training data. This method allows for natural conditioning on reference videos in temporal modules while creating large motion gaps

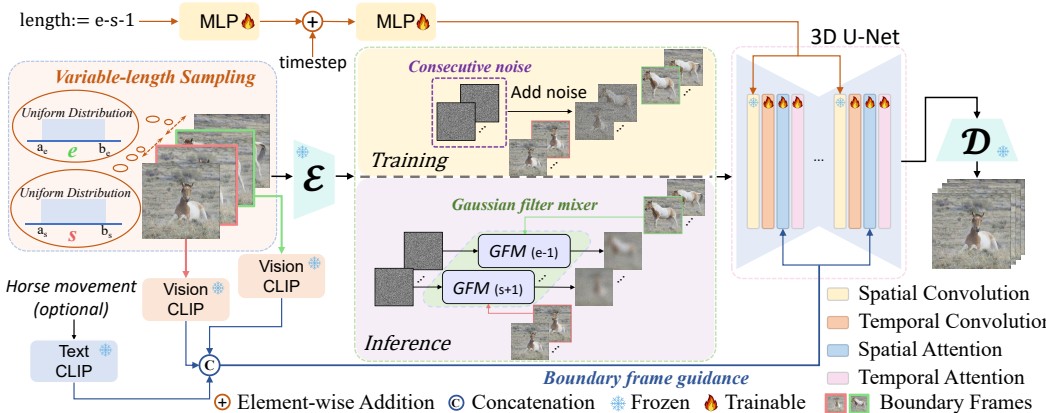

Figure 1: Model architecture. The input sequence is divided into three clips using variable-length sampling. Noise is added exclusively to the latent of the intermediate clip, with length embedded as extra information. Boundary frames are encoded with a CLIP vision encoder as content guidance for spatial transformers. During inference, a Gaussian filter mixer (GFM) is used for noise initialization.

in the middle, enforcing the capture of long-term temporal dependencies. Nevertheless, denoising only the middle part of training data can restrict data utilization and model robustness. To overcome this, we implement a variable-length sampling strategy to optimize data usage and simultaneously enhance the flexibility in generation length.

Furthermore, in text-conditioned generations, text prompts guide the generation direction in spatial modules, where each frame is processed independently, and the temporal modules align them, as vividly illustrated in (Blattmann et al., 2023b). However, in this task, the model operates in a self-supervised fashion and there is no text providing content or semantic guidance to the model. Without such guidance, spatial modules can generate incoherent images, placing extreme burdens on temporal modules to align them. We propose boundary frame guidance (BFG) in spatial modules to mitigate this issue.

Lastly, as revealed in (Lin et al., 2024; Chen, 2023), a train-test noise initialization discrepancy hinders VLDM from generating high-quality videos. While previous solutions in I2V generation (Wu et al., 2023b; Dai et al., 2023) generally involve a shared noise strategy, it is not optimal for this task because the preserved condition frame signal throughout the generation sequence can limit motion range, discouraging synthesis of distinct transition states. To better serve the transition video infilling scenario, we propose a Gaussian filter mixer (GFM) module to balance initialization discrepancy and generation flexibility.

### 3.3 MODEL ARCHITECTURE

The overall model architecture is depicted in Figure 1. In the model training stage, we simulate transition video infilling by dividing a training video into three segments. An entire video sample is first encoded into $z_0 = \{z_0^0, \dots, z_0^{L-1}\}$, and *only the intermediate clip is corrupted* by $t$-step Gaussian noise according to Eq. 2 resulting in $z_t^{\mathcal{I}}$. The input to the U-Net model hence becomes $z_t = \{z_0^{\mathcal{P}}, z_t^{\mathcal{I}}, z_0^{\mathcal{F}}\}$, and the model is optimized to predict $\{\epsilon_t^{s+1}, \dots, \epsilon_t^{e-1}\}$ as per Eq. 3. Loss is computed only on noised frames. Since this approach does not involve extra mask or condition frame concatenation to the channel dimension, it enables the utilization of most pre-trained foundation models that accept 3-channel RGB inputs.

**Variable-length sampling.** To improve data utilization and generation flexibility, we employ variable-length sampling by randomly shifting the start and end points of the infilling clip. Concretely, at each training step, we draw the boundary frame indices $s$ and $e$ randomly from two independent uniform distributions: $s \sim \mathcal{U}(a_s, b_s)$; $e \sim \mathcal{U}(a_e, b_e)$, where $0 < a_s < b_s < a_e < b_e < L - 1$. The resulting length of the noised clip $l := e - s - 1$ thereby follows a triangular distribution with the upper and lower limits $l_{upper} = b_e - a_s$, $l_{lower} = a_e - b_s$. Particularly, when $b_s - a_s = b_e - a_e$, the PDF of the distribution is symmetric and has the mode $l_{mode} = (l_{lower} + l_{upper})/2$. To avoid confusion stemming from variable-length sampling, we equip the model with an awareness of the generation

length it is handling. This is achieved by incorporating a length embedding using sinusoidal encoding followed by an MLP. The length embedding is subsequently added to the timestep encoding and collectively processed through another MLP into the spatial convolution module. This approach improves the model's capacity to leverage training samples by accommodating predictions at varying positions and lengths. It effectively allows for generating at various lengths and accepting reference videos of diverse durations, thereby bolstering the model's robustness and flexibility.

**Dynamic boundary frame guidance.** Boundary frames play a pivotal role in guiding the model's generation as they provide explicit information about the gap the model is tasked to bridge. Therefore, we propose boundary frame guidance (BFG) to compensate for the lack of guidance in transition video generation. Most popular frame conditioning strategies entail extending the keys and values of spatial self-attention layers to include those of the condition frames (Wu et al., 2023a; Ren et al., 2024; Henschel et al., 2024). However, empirical experiments did not prove these approaches effective for this task, and the low-level visual information sometimes restricted the freedom of generation, resulting in synthesized frames copying too much information from existing ones. Instead, we inject the guidance signal into the cross-attention layers via a higher-level CLIP representation. Concretely, we encode the pixel-level boundary frames $x_0^s$ and $x_0^e$ using a CLIP vision encoder and concatenate the representations along the sequence dimension. This integrated representation serves both as content and semantic guidance due to the nature of CLIP representations, informing the model about the generation direction. A short text prompt briefly describing the subject, such as "horse movement", can be optionally leveraged to help classify the action subject or extract knowledge from a pre-trained foundation model. The combined use of the CLIP encoders and the concatenation operation provides the model with a consistent understanding of the integrated condition signal.

**Gaussian filter mixer for inference-time noise initialization.** We propose a dynamic inference-time noise mixing strategy tailored for the transition video infilling task. Since the infilling video functions as a bridge, its first few frames should thereby resemble the preceding video, while the last few frames approach the following video. The frames in the middle should be granted the flexibility to display transition states that are significantly distinct from any reference frames. Inspired by FreeInit (Wu et al., 2023c), we propose a Gaussian filter mixer (GFM) module that dynamically retains a certain amount of information from the closest boundary frame latent. This is accomplished by keeping the low-frequency component of the diffused boundary latent, which offers a rough layout guidance to the denoising process. The preserved information gradually diminishes as the frame position moves away from the boundaries, allowing for greater freedom in generation. It is then mixed with individual Gaussian noise at each frame, resulting in the mixed inference-time noise initialization $\tilde{z}_t^n$ at frame $n$ as

$$\mathcal{F}^{low}(n) = \begin{cases} \mathcal{FFT}_{3D}(z_t^s) \odot \mathcal{G}(f_S(n), f_T(n)) & \text{if } n \leq \dfrac{s+e}{2}, \\ \mathcal{FFT}_{3D}(z_t^e) \odot \mathcal{G}(f_S(n), f_T(n)) & \text{if } n > \dfrac{s+e}{2}, \end{cases} \tag{4}$$

$$\mathcal{F}^{high}(n) = \mathcal{FFT}_{3D}(\epsilon_t^n) \odot (1 - \mathcal{G}(f_S(n), f_T(n))), \tag{5}$$

$$\tilde{z}_t^n = GFM(n) = \mathcal{IFFT}_{3D}(\mathcal{F}^{low}(n) + \mathcal{F}^{high}(n)), \tag{6}$$

where $s$ and $e$ are the indices of the boundary frames; $\mathcal{FFT}_{3D}(\cdot)$ and $\mathcal{IFFT}_{3D}(\cdot)$ represent discrete fast Fourier transform and its inverse operation, performing in 3D dimensions; $f_S(\cdot)$ and $f_T(\cdot)$ are functions that adjust the spatial and temporal stop frequencies, respectively; and $\mathcal{G}(\cdot, \cdot)$ is a 3D Gaussian low-pass filter taking both spatial and temporal stop frequencies as control parameters.

Eq. 4 first ensures that each intermediate frame refers to its closest boundary frame. Subsequently, the adjusting functions progressively reduce the stop frequency values as the distance to the selected boundary increases. We opt for a straightforward linear decreasing function with a scaling coefficient $\lambda$ to realize such control. The stop frequency for both $f_S(\cdot)$ and $f_T(\cdot)$ is computed as

$$f(n) = \max(0, f_0 - \lambda \cdot \min(|n - s|, |n - e|) \cdot f_0), \tag{7}$$

where $f_0$ is the initial stop frequency of the low-pass filter. Here, $f_0$ determines the maximum layout information we aim to retain from the boundary frames, and $\lambda$ regulates the rate at which such information decreases as the synthesis target moves away from the referred boundary.

Table 1: Quantitative comparison with other generative transition models.

| | test-manual | | | | | | test-auto | | | | | |
| | MS-SSIM↑ | PSNR↑ | LPIPS↓ | FVD↓ | CLIP-RS↑ | CLIPSIM↑ | MS-SSIM↑ | PSNR↑ | LPIPS↓ | FVD↓ | CLIP-RS↑ | CLIPSIM↑ |
|---|---|---|---|---|---|---|---|---|---|---|---|---|
| *Horse - 8 frames* | | | | | | | | | | | | |
| DynamiCrafter (Xing et al., 2023) | 0.564 | 17.51 | 0.204 | 802.0 | 0.746 | 0.478 | 0.355 | 15.24 | 0.253 | 355.5 | 0.715 | 0.368 |
| DynamiCrafter-Vid | 0.609 | 17.75 | 0.198 | 1073.6 | 0.830 | 0.468 | 0.430 | 15.96 | 0.264 | 814.2 | 0.790 | 0.380 |
| SEINE (Chen et al., 2023c) | 0.710 | 19.79 | 0.129 | 635.4 | **0.866** | **0.546** | 0.481 | 16.90 | 0.184 | 271.7 | 0.813 | 0.439 |
| SEINE-Vid | 0.588 | 17.54 | 0.179 | 571.7 | 0.719 | 0.480 | 0.434 | 16.23 | 0.224 | 165.6 | 0.700 | 0.395 |
| MAVIN (Ours) | **0.724** | **19.97** | **0.128** | **479.5** | 0.844 | 0.517 | **0.560** | **18.24** | **0.162** | **147.7** | **0.819** | **0.453** |
| *Horse - 12 frames* | | | | | | | | | | | | |
| DynamiCrafter (Xing et al., 2023) | 0.452 | 16.05 | 0.252 | 866.6 | 0.740 | 0.408 | 0.317 | 14.75 | 0.280 | 367.8 | 0.747 | 0.340 |
| DynamiCrafter-Vid | 0.553 | 17.09 | 0.212 | 820.6 | 0.811 | 0.456 | 0.374 | 15.34 | 0.267 | 419.7 | 0.776 | 0.362 |
| SEINE (Chen et al., 2023c) | 0.591 | 17.78 | 0.176 | 743.6 | 0.815 | 0.487 | 0.383 | 15.48 | 0.236 | 289.6 | 0.782 | 0.379 |
| SEINE-Vid | 0.357 | 14.15 | 0.330 | 1096.0 | 0.522 | 0.349 | 0.283 | 13.97 | 0.327 | 343.1 | 0.570 | 0.294 |
| MAVIN (Ours) | **0.666** | **19.12** | **0.148** | **559.9** | **0.844** | **0.491** | **0.458** | **16.68** | **0.211** | **208.8** | **0.798** | **0.400** |
| *Tiger - 8 frames* | | | | | | | | | | | | |
| DynamiCrafter (Xing et al., 2023) | 0.383 | 15.58 | 0.251 | 856.8 | 0.733 | 0.447 | 0.260 | 13.81 | 0.309 | 377.9 | 0.691 | 0.415 |
| DynamiCrafter-Vid | 0.477 | 16.35 | 0.224 | 1177.3 | 0.822 | 0.477 | 0.408 | 15.22 | 0.239 | 619.5 | 0.836 | 0.491 |
| SEINE (Chen et al., 2023c) | 0.613 | 18.23 | 0.152 | 612.5 | **0.860** | **0.553** | 0.417 | 15.29 | 0.211 | 297.3 | 0.844 | 0.506 |
| SEINE-Vid | 0.580 | 17.83 | 0.177 | **447.5** | 0.766 | 0.528 | 0.525 | 16.49 | 0.182 | **232.7** | 0.798 | 0.544 |
| MAVIN (Ours) | **0.678** | **19.17** | **0.137** | 536.8 | 0.846 | 0.530 | **0.635** | **17.87** | **0.139** | 245.3 | **0.869** | **0.562** |
| *Tiger - 12 frames* | | | | | | | | | | | | |
| DynamiCrafter (Xing et al., 2023) | 0.346 | 15.13 | 0.276 | 834.3 | 0.763 | 0.423 | 0.221 | 13.50 | 0.336 | 395.7 | 0.744 | 0.385 |
| DynamiCrafter-Vid | 0.396 | 15.51 | 0.253 | 873.1 | 0.793 | 0.443 | 0.290 | 14.23 | 0.288 | 371.5 | 0.803 | 0.434 |
| SEINE (Chen et al., 2023c) | 0.504 | 16.86 | 0.196 | 707.4 | **0.859** | **0.500** | 0.361 | 14.61 | 0.245 | 356.3 | 0.847 | 0.472 |
| SEINE-Vid | 0.390 | 15.49 | 0.268 | 733.5 | 0.703 | 0.425 | 0.277 | 13.72 | 0.314 | 423.6 | 0.675 | 0.405 |
| MAVIN (Ours) | **0.595** | **18.08** | **0.167** | **689.7** | 0.835 | **0.500** | **0.513** | **16.23** | **0.183** | **310.9** | **0.852** | **0.512** |

# 4 EXPERIMENTS

## 4.1 EXPERIMENTAL SETUP

**Datasets.** For our experiments, we focus on two distinct animal species to verify the effectiveness of the proposed method: horses and tigers. We use the AnimalKingdom dataset (Ng et al., 2022) for training the horse model and the TigDog dataset (Del Pero et al., 2015) for the tiger model. The AnimalKingdom dataset encompasses a diverse range of species and tasks, but we only utilize the videos labeled as "Horse" from the *action_recognition* task for training. However, we noticed that the video clips in *action_recognition* are generally too short and correspond to only single actions, resulting in insufficient action transition patterns for model training. Therefore, we supplemented the horse training data with additional long-take web videos that capture horse movements. Consequently, the total duration of the training data for each dataset is approximately 45 minutes under 30 FPS.

Testing clips for the transition video infilling task should ideally contain action transitions or large motions to effectively evaluate the model's efficacy. Such data, however, is challenging to source from existing datasets, prompting us to construct our own. We collect videos from the Internet and generate the test data in two ways: manual cutting, which yields high-quality samples, and automatic generation, which produces a large number of test clips. We refer to the test sets generated in these ways as *test-manual* and *test-auto*, respectively. For *test-manual*, we meticulously cut web videos into 32-frame clips by ensuring the occurrence of significant movements or posture changes (e.g., transitioning from grazing to standing upright) in the intermediate clips. We curated 34 such test samples for each animal class. For *test-auto*, we employ an optical flow estimator, RAFT (Teed & Deng, 2020), to estimate the motion intensity between the two reference clips. Concretely, we select video clips based on the average optical flow magnitudes of the boundary frames. Since small magnitude values suggest minor motions and excessively large values typically result from dramatic camera movements, only those with values falling within a certain range are leveraged. To formalize this, the boundary frame indices $s$ and $e$ for *test-auto* are selected using the following equation:

$$\{(s,e)\} = \left\{ (s,e) \mid \frac{1}{h \cdot w} \sum_{i=1}^{h} \sum_{j=1}^{w} \|\mathcal{E}_{flow}(x_s, x_e)_{i,j}\|_2 \in (T_{lower}, T_{upper}), e - s - 1 = l_{test} \right\}, \quad (8)$$

where $T_{lower}$ and $T_{upper}$ are the lower and upper thresholds; $h$ and $w$ are the height and width of estimated optical flows; and $l_{test}$ is the length of the generation sequence we want to test. We tuned the thresholds and obtained 113 visually satisfactory test samples for the horse class and 104 for the tiger class by setting $l_{test}$ to 12.

**Implementation details.** We initialize our model from ModelScopeT2V-1.7b (Wang et al., 2023a) and fine-tune it with the proposed framework for 40K steps. The optimization is carried out using an AdamW optimizer (Loshchilov & Hutter, 2017), with a constant learning rate of 5e-6 and a batch size of 1. Training videos are randomly sampled into 32-frame clips at a sample rate of 2,

and pre-processed to eliminate potential shot transitions by excluding clips where any SSIM (Wang et al., 2004) value between consecutive frames falls below 0.1. Videos shorter than 32 frames are discarded. Experiments are conducted at a resolution of 256×256. Training and inference require around 40 and 12 GB vRAM, respectively. All training is performed on a single NVIDIA L40 GPU, with each trial taking approximately one day. The length range of random intermediate clips is set to $l_{lower} = 2, l_{upper} = 22$. The GFM parameters employed are $f_0 = 0.6, \lambda = 0.1$.

**Evaluation Metrics.** We present the following evaluation metrics: multi-scale structural similarity (MS-SSIM)(Wang et al., 2003), peak signal-to-noise ratio (PSNR), LPIPS(Zhang et al., 2018), FVD (Unterthiner et al., 2019), and CLIP similarity (Radford et al., 2021). However, most of these metrics primarily assess reconstruction quality and similarity between generated and ground-truth videos, considering each frame independently. They do not adequately account for temporal coherence, which reflects the smoothness of generated motions. To address this gap, we propose a CLIP-similarity-based inner-frame consistency measurement to quantify the *relative smoothness* with respect to the ground truth video clip. We term this measure CLIP Relative Smoothness score (CLIP-RS), computed as follows:

$$\text{CLIP-RS} = \frac{1}{L-1} \sum_{i=1}^{L-1} \frac{\min(\text{CLIPSIM}(p_{i-1}, p_i), \text{CLIPSIM}(q_{i-1}, q_i))}{\max(\text{CLIPSIM}(p_{i-1}, p_i), \text{CLIPSIM}(q_{i-1}, q_i))}, \tag{9}$$

where $p_i$ is the $i$-th generated frame and $q_i$ is the corresponding ground truth frame. $L$ is the length of the generated video. $\text{CLIPSIM}(p_i, p_j)$ denotes the cosine similarity between the CLIP representations of images $p_i$ and $p_j$. Each summation term quantifies the relative frame change in the generated video compared to the actual change in the ground truth. Either a relatively drastic or subtle change results in a low score. For example, if the oracle transition occurs at a steady rate while the synthesized video initially remains stationary and then abruptly changes to complete the transition, the differences in transition pace will be captured, leading to low relative smoothness values.

CLIP-RS is a metric calculated along the frame axis, measuring the degree of changes between adjacent frames. Although it references the ground truth video, it does not engage in any direct frame-to-frame comparisons between the two videos. This characteristic renders this metric indifferent to the quality of the generated images or their resemblance to the original video.

Table 2: CLIP-RS responds to temporal changes and is not sensitive to visual aesthetics.

| | SSIM↑ | PSNR↑ | LPIPS↓ | CLIPSIM↑ | **CLIP-RS↑** |
|---|---|---|---|---|---|
| Original (self-comparison) | 1.00 | inf | 0.00 | 1.00 | 1.00 |
| Decrease luminance by 50% | 0.71 | 13.6 | 0.21 | 0.66 | 0.97 |
| Increase contrast by 50% | 0.69 | 19.9 | 0.08 | 0.82 | 0.97 |
| Zeroing-out red channel | 0.67 | 11.3 | 0.29 | 0.66 | 0.96 |
| Zeroing-out red&green channels | 0.33 | 8.44 | 0.60 | 0.49 | 0.95 |
| Replicating 1st frame as video | 0.79 | 20.7 | 0.06 | 0.77 | 0.87 |

their resemblance to the original video. We demonstrate this by manipulating a 12-frame video clip and computing the metrics with the original video. As shown in Table 2, when the video's visual aesthetics are perturbed (rows 2-5), metrics based on predicted-actual similarity are significantly impacted despite the structural content and motion effect of the video remaining unchanged, whereas CLIP-RS maintains a score close to 1. In contrast, when the video's temporal property is altered (the last row, where the new video is comprised of a 12-time repetition of the original video's first frame), similarity-based and quality-based metrics yield superior results compared to when visual aesthetics were disturbed, while CLIP-RS can identify such smoothness discrepancies. This is in direct contrast to the use of absolute smoothness measurement Chen et al. (2023c), where a static video can achieve a perfect smoothness score of 1, which contradicts our objective. The $\text{CLIPSIM}(\cdot, \cdot)$ function in CLIP-RS can also be substituted with other similarity measurements such as SSIM, averaged optical flow momentum, etc.

## 4.2 RESULTS

**Comparison with existing methods.** For our comparative analysis, we selected two open-source diffusion-based generative models: DynamicCrafter (Xing et al., 2023) and SEINE (Chen et al., 2023c). Both models are capable of generating transition videos from two condition images. However, to ensure a more fair and relevant comparison to our work, we also conducted experiments where these models were conditioned on video inputs. We refer to these modified versions as DynamicCrafter-Vid and SEINE-Vid.

We conducted experiments with two infilling length settings: (i) generating 8 frames given 12-frame condition clips on each side, and (ii) generating 12 frames given 10-frame references. The total input

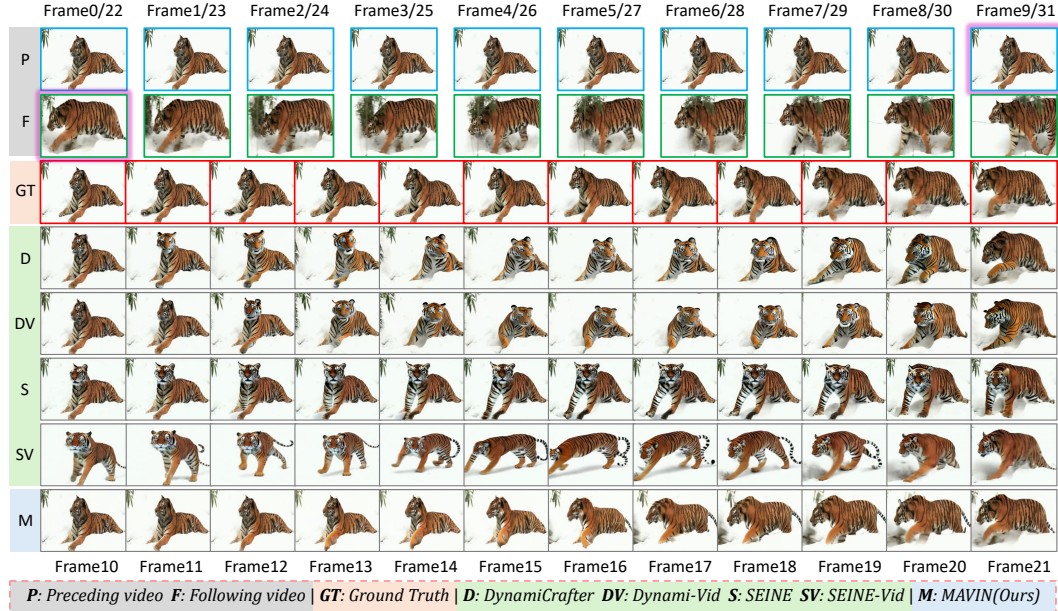

Figure 2: Qualitative comparison of MAVIN with baseline models. The top two rows are input reference videos, with glowing frames marking the boundaries. MAVIN demonstrates smoother and more natural transitions and superior spatiotemporal consistency compared to baseline models.

length is 32, matching our test set samples. Particularly, we found that DynamicCrafter, which was trained to generate fixed 16-frame videos, performed poorly when this length was altered. Therefore, for DynamicCrafter, we use a 4-frame reference on each side for 8-frame infilling, and 2 for 12-frame infilling, maintaining a total length of 16. For image-conditioned generations, where DynamicCrafter generates 14 frames (16 minus 2 reference images), we evenly sampled 8 and 12 frames from the 14 for metric computations. All metrics were calculated only on intermediate clips, except for FVD, which was compared with the entire input sequence.

We show quantitative results in Table 1 and qualitative results in Figure 2. MAVIN substantially outperforms other generative baseline methods, especially when the motion is difficult. Specifically, SEINE is the most competitive transition generation model, but as the number of infilling frames increases, the gap between MAVIN and SEINE becomes obvious. *test-auto* is generated at a sample rate of 4, which is rather challenging. It is equivalent to bridging a 48-frame gap when infilling 12 frames on *test-auto*. The performance gap further increases under this setting, showing the effectiveness of the proposed method in infilling videos with large and complex motions.

As the only existing generative model trained for transition purposes, SEINE adopts a BERT-like masking strategy for masked modeling, where each frame is corrupted by chance independently, resulting in an intermittent corruption pattern. Although this method enhances data utilization and robustness, it falls short in generating long-term temporally cohesive videos because the corruption pattern allows the model to rely on nearby clean frames for predictions. In contrast, our method consistently applies continuous corruption up to a maximum length of 22 frames, compelling the model to capture long-term motion dependencies.

**Ablation Study.** We ablate the two key components of the proposed framework and present the qualitative results in Table 3. Results were obtained on *Horse test-manual* by predicting 12 frames under the same experimental setup. Boundary frame guidance (BFG) offers important content direction during model training, and the Gaussian filter mixer (GFM) helps stabilize the generation by providing essential information to address the discrepancy between training and inference phases. By ablating either BFG or GFM, the performance deteriorates across all metrics. When both components are removed together, the model experiences severe degradation, demonstrating the effectiveness and necessity of these components for high-quality video generation. (See supplementary materials)

Table 3: Ablation study on boundary frame guidance (BFG) during training and Gaussian filter mixer (GFM) noise initialization during inference.

| | MS-SSIM↑ | PSNR↑ | LPIPS↓ | FVD↓ | CLIP-RS↑ | CLIPSIM↑ |
|---|---|---|---|---|---|---|
| MAVIN (Proposed Method) | 0.666 | 19.12 | 0.148 | 559.9 | 0.844 | 0.491 |
| −Boundary Frame Guidance | 0.651 (-2.3%) | 19.00 (-0.6%) | 0.153 (-3.4%) | 570.9 (-2.0%) | 0.833 (-1.3%) | 0.483 (-1.6%) |
| −Gaussian Filter Mixer | 0.647 (-2.9%) | 18.03 (-5.7%) | 0.167 (-12.8%) | 627.2 (-12.0%) | 0.815 (-3.4%) | 0.475 (-3.3%) |
| −BFG −GFM | 0.606 (-9.0%) | 17.78 (-7.0%) | 0.189 (-27.7%) | 672.2 (-20.1%) | 0.781 (-7.5%) | 0.443 (-9.8%) |

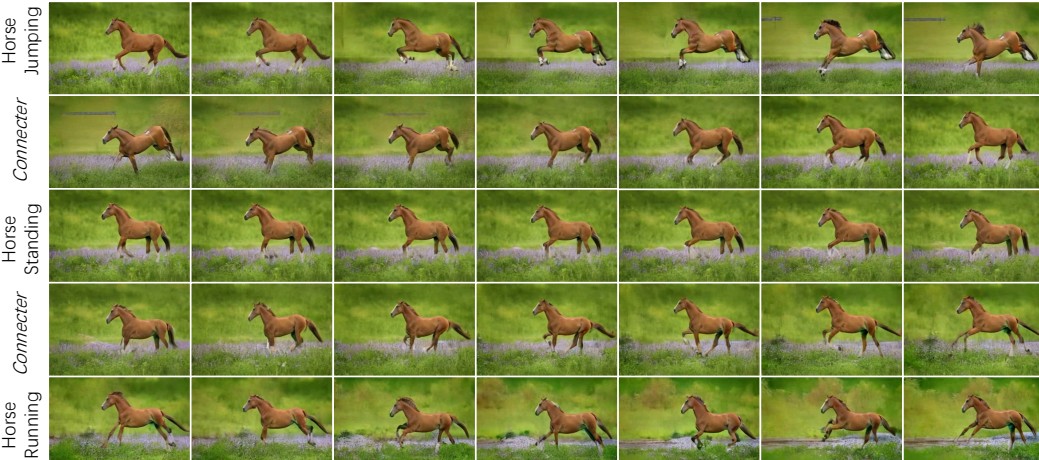

*A horse jumps first, then stands still, and finally runs.*

Figure 3: Application of the transition video infilling model. It connects multiple TI2V-generated single-action video clips into a cohesive extended video with smooth and natural action transitions.

## 4.3 Application for Multi-Action Generation

We achieve multi-action generations by connecting single-action videos with MAVIN. This work does not focus on optimizing the single-action models; instead, we employ existing TI2V models that animate an input image through text control, as discussed in Section 2. In our empirical experiments, directly generating large motions or non-continuous actions using pre-trained TI2V models (Dai et al., 2023; Ren et al., 2024) led to failure. Therefore, we fine-tune these models for the single-action generation purpose. To integrate the synthesized single-action videos, we insert noise of the desired length between two videos and use MAVIN to infill a transition video. Alternatively, instead of inserting noise, we can concatenate single-action videos and replace the junction frames with noise to regenerate the transition parts.

We initialize the action model from AnimateAnything (Dai et al., 2023). The training data for single actions is also derived from the AnimalKingdom and TigDog datasets, except that we collect additional data for training the action "horse jumping". We train one model per animal species. A fixed action prompt, such as "horse is jumping", is tied to each action and serves as the text condition during training.

To create a multi-action video, we first use a single image, controlled by multiple action prompts, to generate multiple single-action videos separately. These videos are then concatenated into a longer sequence, arranged in the desired order. We regenerate the junction frames using the infilling model for smooth action transitions. Figure 3 illustrates an example of such an application. In this example, we generate 20 frames for each single action and refine 12 frames centered around each junction, resulting in a 60-frame-long video that contains three actions: jump, stand, and run. The first, third, and fifth rows depict the single-action videos generated by the action model, while the second and fourth rows are generated by the infilling model to connect them. This approach generates highly temporally cohesive examples with great flexibility.

## 5 CONCLUSION

In conclusion, this study has presented a novel approach to generative video infilling, specifically targeting the generation of transition clips in multi-action sequences, leveraging the capabilities of diffusion models. Our model, MAVIN, demonstrates a significant improvement over existing methods by generating smoother and more natural transition videos across complex motion sequences. This research lays the groundwork for future advancements in unsupervised motion pre-training, large-motion video interpolation, and multi-action video generation. While this technique enables new applications, it is also crucial to establish guidelines and implement safeguards to prevent its potential misuse in creating fake content, which raises ethical and security concerns.

**Limitations.** Due to computational limitations and the proprietary nature of the widely used video training dataset WebVid-10M (Bain et al., 2021), our experiments were conducted only under specific scenarios and initialized from existing foundation models. Further exploration of the task might require training at scale. Moreover, while we did not concentrate on optimizing the single-action (TI2V) models, a notable trade-off between visual quality and motion intensity persists even after fine-tuning, highlighting an area for further research. The failure cases include the single-action model's inability to follow the action prompt and the inconsistency in appearance in later frames for actions involving large motions.

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
