# OpenReview forum: "MAVIN: Multi-Action Video Generation with Diffusion Models via Transition Video Infilling"
_ICLR.cc/2025/Conference — Submitted to ICLR 2025_

### Official Review · Reviewer_23Cr · 2024-10-17

**Soundness:** 3
**Presentation:** 3
**Contribution:** 2
**Rating:** 5
**Confidence:** 4

**Summary:**

This paper proposes a video prediction method that outputs intermediate transition frames given varying numbers of preceding and succeeding frames. Additionally, a CLIP-based evaluation method is introduced to verify inter-frame coherence.

**Strengths:**

1. The MAVIN model is proposed to generate transition videos between two independent video segments, and it is capable of performing this task.

2. BFG is introduced to guide the Diffusion process in generating transition videos, and GFM is proposed to smooth noise for better quality generation.

3. A new metric, CLIP-RS, is introduced to evaluate whether the smoothness of generated videos aligns with the original videos.

4. The paper presents clear illustrations, with Figure 1 effectively conveying the entire model's flow.

5. The authors provide code to enhance reproducibility.

6. The method has also been applied in extended contexts.

**Weaknesses:**

1. This task has some overlap with video prediction or frame interpolation, making it not particularly novel, and there are numerous models that have completed this work.

2. I believe the effectiveness of the proposed method lies in the use of Dynamic Boundary Frame guidance, which is fundamentally about injecting image features into cross-attention using CLIP—this technique has been widely applied in Image to Video generation.

3. Regarding GFM, many papers have discussed the influence of initialization noise on generation [1].

4. The testing data in this paper is somewhat limited; for example, it only uses two self-created datasets and does not test on publicly available datasets.

5. For the ICLR conference, this paper lacks explanatory depth.

[1] Wu T, Si C, Jiang Y, et al. Freeinit: Bridging initialization gap in video diffusion models[J]. arXiv preprint arXiv:2312.07537, 2023.

**Questions:**

1. What does "PDF" mean in line 215?

2. In line 427, the roles of BFG and GFM are discussed, but the data in Table 3 does not convincingly support this viewpoint. For instance, while BFG provides contextual guidance, removing BFG does not result in significant changes to the Clipscore; only when both are removed do substantial changes occur. Does this prove the combined effect of BFG and GFM?

3. There are many ways to transition between the two input videos; is it reasonable to use SSIM to evaluate the structural similarity of generated videos and GT frame by frame? For example, in Figure 2, it is reasonable for the tiger to rise in any frame, but using SSIM requires the tiger to rise at a specific frame.

---

> ### Author Response · Authors · 2024-11-21
>
> Thank you for your valuable and constructive feedback. We would like to respond to your concerns individually as follows:
>
> #### Weaknesses
> 1. Despite being similar to video interpolation, they address different tasks and scenarios. The major difference in our transition infilling task is that the infilling span is long (16 frames), and its content is uncertain and ambiguous. Thus, it cannot be determined by calculation-based algorithms. Video interpolation algorithms focus on limited and unambiguous contents in between adjacent frames, and are not tasked to interpolate an entire video sequence with large motion gaps. In the AIGC era, this task is newly made possible by generative models. And compared to video prediction. The proposed task is bi-directional conditioned. The generated content should not only consider the preceding content but also how to smoothly transit into the following one. The additional constraint also makes it more difficult than causal prediction.
> 2. The technique itself is not a new one, but it stands as a purpose-driven solution tailored to address the unique challenges of our task. Although various existing methodologies could be adapted, we found the BFG approach particularly essential for this specific scenario. Our strategy of encoding and concatenating the two edge frames as a guidance signal is also a deliberate design choice that mitigates the lack of guidance encountered during unsupervised training (where no textual description of the target action is provided).
> 3. We borrow the idea and make adaptations to our transition-infilling task.
> 4. We recognize the limitations of our work. Resource constraints and the lack of suitable existing datasets have hindered our ability to scale up at this time. Nonetheless, we have taken the crucial first step of proposing the task and validating the feasibility of our approach. We believe that multi-action video generation will and needs to be paid more attention in the future. While pioneering this direction and proposing a perfect solution is impracticable, we hope such a bi-directional unsupervised training pipeline can be valued and spur further expansion on this topic.
> #### Questions
> 1. "PDF" in line 215 refers to the probability density function.
> 2. Since the effect of GFM is rather strong, when it is applied, the gap of w/ and w/o BFG is mitigated. However, if GFM is not applied, removing BFG results in major performance degradation (please see the last two rows of the ablation results). In other words, adding BFG during training or not has big differences.
> 3. We understand the concern. For long durations, there will be problems as you described. However, the duration of the infilling target in the test data is not that long and the transition basically happens throughout the intermediate clip. Especially for the test-manual, which is hand-made to ensure so. Within the limited infilling duration (16 frames) if the tiger rises too rapidly (e.g. using only 4 frames) it will seem unnatural and also lead to a low CLIP-RS score. Otherwise, the model and gt are both trying to perform the same action through the entire target clip; a little out-sync in this case is totally normal and does not affect the evaluation too much. Baseline methods either perform too many unnecessary actions or generate irrelevant actions/contents, which can be captured by the gt-based metrics.

---

> > ### Comment · Reviewer_23Cr · 2024-11-26
> >
> > Thank you, your answer solved my doubts. But I think the difference between this job and previous jobs is still not obvious. So I decided to keep the score.

---

### Official Review · Reviewer_zH8s · 2024-10-31

**Soundness:** 3
**Presentation:** 3
**Contribution:** 2
**Rating:** 5
**Confidence:** 4

**Summary:**

The paper addresses the task of generating smooth and natural transitions between two video segments. A consecutive denoising strategy with variable-length sampling is adopted for infilling training. Boundary frame guidance is proposed to ensure semantically coherent frames, and the Gaussian filter mixer method is applied to balance the noise discrepancy between training and testing. A novel CLIP-relative smoothness metric is introduced to evaluate temporal coherence and smoothness. Experimental results on horse and tiger videos show that the proposed method achieves superior performance compared to SEINE and DynamiCrafter.

**Strengths:**

- Among the proposed components, the GFM module is an especially interesting idea.
- The authors specifically introduced the CLIP-RS metric to quantify relative smoothness, and Table 2 provides valuable insights on this metric.
- The videos in the supplementary materials and results in Table 1 demonstrate that MAVIN achieves superior results compared to prior work using the proposed benchmark.
- The ablation study effectively shows the impact of the proposed BFG and GFM components.

**Weaknesses:**

- The dataset used in the experiments is limited to tiger and horse videos, restricted domains with only 45 minutes of video available for training. Furthermore, the method requires fine-tuning an existing T2V model for 40k steps with a batch size of 1. This setup is limited in both scope and scale, making it insufficient to fully demonstrate the proposed method’s effectiveness. Larger-scale experiments across multiple domains would strengthen the evaluation.
- The compared models, SEINE and DynamiCrafter, are trained on open-domain data, enabling them to handle a broader range of scenarios. However, they are not specifically trained on animals, which makes the comparison to MAVIN (focused on animal videos) less direct. Training SEINE and DynamiCrafter on the proposed benchmark could enable a fairer comparison.
- The proposed components rely on standard approaches in video generation. For example: 1) the infilling objective in L201–L208, 2) variable-length sampling with a length embedding added to the timestep encoding, and 3) BFG uses CLIP embedding rather than a low-level embedding. These contributions, while effective, are not technically new.

**Questions:**

The authors argue that the BERT-like training in Chen et al. is less effective due to the alternating appearance of target and reference frames. To address this, they propose applying noise to a consecutive subsequence of training data. However, this approach is relatively standard, as seen in methods like MAGVIT (Yu et al.). Providing additional background and discussion would help contextualize this choice and highlight its novelty.

---

> ### Author Response · Authors · 2024-11-21
>
> Thank you for your valuable and constructive feedback. We would like to respond to your concerns individually as follows:
> #### Weaknesses
> 1. We recognize the limitations of our work. Resource constraints and the lack of suitable existing datasets have hindered our ability to scale up at this time. Nonetheless, we have taken the crucial first step of proposing the task and validating the feasibility of our approach. We believe that multi-action video generation will and needs to be paid more attention in the future. While pioneering this direction and proposing a perfect solution is impracticable, we hope such a bi-directional unsupervised training pipeline can be valued and spur further expansion on this topic.
> 2. We understand the concerns regarding the baseline comparison. However, making a totally fair comparison is rather impractical. As highlighted, SEINE and DynamiCrafter are large-scale, general-purpose, and data-heavy models, while MAVIN is designed to be lightweight, task-specific, and data-efficient. In such a case, further fine-tuning SEINE and DynamiCrafter on our animal-specific benchmark could indeed lead to a more unfair condition, because their models are already extensively pre-trained on vast amounts of data for transition tasks.  If we further fine-tune them on the WHOLE set of our training data,  the overwhelming pre-training and data superiority could make the methodology layer negligible.
>     Instead, our current comparison neutralizes their model's generality aspect with the fact that our model is low-resource trained on less than 1-hour total video duration, whereas theirs is about 52,000 hours in total. Considering the scope differences, the low-resource task-specific setup makes the comparison relatively more fair.
>     And from a more practical and realistic aspect, they did not open-source the training code but only inference scripts, nor do we have the resources to fine-tune a foundation model at that scale.
> 3. While these ideas themselves are not new in the field, they are purpose-driven and deliberately designed for the proposed task. They are necessary components that we explored aiming to make the approach work, rather than finding innovational improvement on existing methods.
> #### Question
> Most current interpolation and prediction models adopt the BERT-like approach, which works well for predicting short missing content. However, for our task, we need to generate entire segments of video while considering the content and coherence of both preceding and succeeding video segments. The BERT approach makes the model prone to relying on the nearest unmasked tokens to generate content, thus failing to train the model's ability to handle larger gaps without reference. In the prediction stage, when the model needs to independently generate 16 consecutive frames without reference, it becomes inefficient. This can be observed from the baseline model Seine's generation performance; once the number of infilling frames increases, its generated content cannot maintain consistency and relevance.
> In our approach, masking a consecutive subsequence of training data forces the model to acquire the ability to generate coherent long-range content, thereby making the content-ambiguous transition generation possible. We appreciate the suggestion for additional background and will ensure to include a more comprehensive discussion to better contextualize our method and highlight its novelty.

---

> > ### Comment · Reviewer_zH8s · 2024-11-27
> >
> > Thank you to the authors for their response. The discussion on W2 and Q1 has addressed some of my questions. However, my main concern remains the limited training and test data used in the experiments, which was also noted by the other three reviewers. Therefore, I have decided to maintain my rating.

---

### Official Review · Reviewer_zNvk · 2024-11-03

**Soundness:** 3
**Presentation:** 3
**Contribution:** 3
**Rating:** 6
**Confidence:** 4

**Summary:**

This video proposes a method for filling in between two videos to build multi-action videos, while most video generation models are focused on generating videos containing one atomic action. For this goal, a training paradigm has been proposed to ensure the smoothness and preserving context in the transition state of a video. Authors also proposed a new evaluation metric.

**Strengths:**

The proposed method has enough novelty.
The authors touched on the problem of evaluation of the multi-action video generation.

**Weaknesses:**

The main weakness of this manuscript is the small scope of its train/test data. Also, the few examples make it very hard to draw a conclusion.

**Questions:**

1- MAVIN performs relatively better under the `test-auto` compared to the baselines. What could be the reason?
2- I don't have a clear understanding of how Eq.5 & 6 outputs are being used. Can you define each term and also explain more of where each is being used? Also, it would be beneficial if their usage be visualized in Fig.1.
3- Why not use a simple (like linear) interpolation between (z^s and z^e)? How much performance advantage is gained by the FFT and IFFT operations and doing the low pass filtering in the frequency domain?
4- Motion (and its smoothness) is very important in temporal coherence. How CLIP-RS address that? If the CLIP-RS can just measure the contextual coherence and not the motion, which other metric do you observe for that?

---

> ### Author Response · Authors · 2024-11-21
>
> Thank you for your valuable and constructive feedback. We would like to respond to your concerns individually as follows:
>
> #### Weakness
> We recognize the limitations of our work. Resource constraints and the lack of suitable existing datasets have hindered our ability to scale up at this time. Nonetheless, we have taken the crucial first step of proposing the task and validating the feasibility of our approach. We believe that multi-action video generation will and needs to be paid more attention in the future. While pioneering this direction and proposing a perfect solution is impracticable, we hope such a bi-directional unsupervised training pipeline can be valued and spur further expansion on this topic.
> #### Questions
> 1. The *test-auto* samples one frame from every four frames in the original video, resulting in the whole video spanning four seconds. The motion range of the video is therefore larger and more challenging. MAVIN achieves more noticeable performance superiority to baseline methods under this setup due to the increased complexity.
> 2. Eq5 and Eq6 are used for inference noise initialization. The blurred images in Fig1 correspond to the Eq6 output.
> 3. In fact, we had tried this method in early experiments. Not only did it fail to bring improvement, but it also resulted in a decline. Mixing two edge images featuring two different action states leads to a naive mixture that corrupts either image's signal, therefore as a noise initialization, it may confuse the model. Instead, we use either one of the edge images (the nearest) and adjust the residual of it. We found this approach to be working great. As for the comparison of keeping the low-frequency band vs the whole frequency band, we suggest referring to the FreeInit paper where detailed analyses and comparisons were given.
> 4. CLIP-RS can be used with any similarity-based metric, such as optical-flow-RS. However, currently, there is no particularly effective method to characterize motion smoothness effectively; it is mainly observed visually. The naturalness and smoothness difference between baseline modes and MAVIN, when assessed by eyes, was quite obvious. The baseline methods show noticeable abruptness and inconsistency. Please refer to the video examples in the supplementary material to see the comparison.

---

> > ### Comment · Reviewer_zNvk · 2024-11-26
> > **More clarifications for Q1**
> >
> > Do you mean that for the `test-manual`, the motion range of the videos is shorter (compared to `test-auto`), and that's why your model is outperforming the baselines more? In that case, do you believe that the longer the motion-range is, the more your model shine compared to other baselines?

---

> > > ### Author Response · Authors · 2024-11-27
> > >
> > > Yes, that's correct. The `test-manual` typically samples one frame every two frames, while `test-auto` samples one per four frames. Accordingly, the 8-frame generation testing on the `test-manual` set would have a motion range spanning **16** (8\*2) frames in the original video, and the 12-frame setup spanning **24** (12\*2). For `test-auto`, these two setups would span **32** (8\*4) and **48** (12\*4) frames respectively.  The gains compared to SEINE under these setups are about 2%, 12.7%, 16.7%, and 21.1%, respectively, if evaluated on horses with MS-SSIM for example. This corresponds to the conclusion that the greater the motion range is, the more MAVIN outperforms baseline methods.

---

### Official Review · Reviewer_UVGm · 2024-11-04

**Soundness:** 3
**Presentation:** 3
**Contribution:** 3
**Rating:** 5
**Confidence:** 2

**Summary:**

This paper proposes a diffusion-model based method to generate transition video frames for two given videos. It first designs a consecutive noising strategy coupled with variable-length sampling for handling large infilling gaps and varied generation lengths. To address the lack of semantic guidance during transition generation, it utilizes boundary frames as guidance. During inference, Gaussian filter mixer (GFM) dynamically manages noise initialization, mitigating train-test discrepancy while preserving generation flexibility. Finally, it introduces a new metric, CLIP-RS (CLIP Relative Smoothness), to evaluate temporal coherence and smoothness. Experiments are conducted on two animal datasets and demonstrate good results.

**Strengths:**

1. The motivation of proposed methods is clear introduced.
2. The paper writing is good and easy to follow.

**Weaknesses:**

1. It needs a clear definition for multiple actions in the Abstract and Introduction. And how existing method processes this scenario.
2. In the related work, please state how different between the proposed method and existing works, which helps reader to decide the novelty.
3. Are the two datasets are multiple action videos?
4. More quality results are needed and it is better to show some failure cases.

**Questions:**

See the above weaknesses.

---

> ### Author Response · Authors · 2024-11-21
>
> Thank you for your valuable and constructive feedback. We would like to respond to your concerns individually as follows:
>
> 1. Multi-action generation is relatively understudied, and existing works all failed to consider this aspect. Our proposed method distinguishes itself by decomposing multi-action generation into a single-action generation and a connection-infilling task, with the latter being a novel task that has not been explored. We will state it more clearly in the revision.
>
> 2. The related works we reviewed include video generation, video interpolation, and scene transition. Despite their relevance, none of them address the same problem we are solving. For instance, video interpolation fails to infill consecutively long and uncertain video content, video prediction is not conditioned on bi-directional constraints, and scene transition only considers image borders, neglecting temporal coherence constraints in video transition. We will also improve the Introduction and Related Work sections to more clearly convey these core ideas.
>
> 3. The manual data ensures action transitions, considering at least two distinct actions. On the other hand, the automatic data, filtered via optical flow thresholds, selects significant movements but does not always ensure two distinct actions.
>
> 4. The supplementary material includes more examples. We will analyze and select some most representative failure examples to include together in the appendix in our next modified version to better illustrate our strengths and limitations. We appreciate this valuable and insightful suggestion.

---

### Official Review · Reviewer_4GmR · 2024-11-04

**Soundness:** 2
**Presentation:** 4
**Contribution:** 2
**Rating:** 6
**Confidence:** 4

**Summary:**

This paper introduces MAVIN, a framework for the transition video infilling task, which aims to generate an intermediate video clip between two adjoining clips, ensuring a fluid and seamless transition. To enhance the performance of the video model, the authors propose several methods. First, they introduce a self-supervised training pipeline that allows transition video infilling without the need for text prompt guidance. To address large infilling gaps and accommodate varied generation lengths, they employ a consecutive noising strategy combined with variable-length sampling. Second, boundary frame guidance is utilized to mitigate issues caused by the lack of semantic guidance. Third, a Gaussian Filter Mixer is proposed to balance initialization discrepancies and generation flexibility. Additionally, they introduce a metric to evaluate temporal coherence and smoothness. This approach is significant for its potential in fields like digital art making, where smooth and natural transitions are crucial. Extensive qualitative experiments demonstrate the effectiveness of the proposed method. However, I have concerns regarding the experimental setup (see weaknesses section).

**Strengths:**

This paper focuses on the transition video infilling with diffusion models, which is an important research topic that is relatively under-explored compared to the Text-to-Video or Video-to-Video setting. It is significant due to its potential applications in fields like digital art making. Generating a transition video with large motion gaps while maintaining motion consistency  is a challenge problem, for which this paper contributes interesting training pipeline and system design for offering guided information and stabilizing the inference. In particular, the use of GFM to balance initialization discrepancy and generation flexibility is technically sound and interesting. The generated results seem to be more smooth and natural than other methods by leveraging the proposed methods.

**Weaknesses:**

Lack of discussion and comparison with most relevant works on the  transition video infilling setting (e.g., [1], [2]).
Lack of ablation studies.
The experiment results on horses and tigers data set is not convinced, since it is too simple. I recommend the author to examine the proposed method on open-domain test set. I also expect the video result on digital art or cartoon scenarios.

[1] Generative Inbetweening: Adapting Image-to-Video Models for Keyframe Interpolation, 2024.

[2]  Explorative inbetweening of time and space, 2024.

Overall, while I appreciate the important problem that this paper proposes to solve. However, the unsatisfactory performance could come from pre-mature experiments. I believe this paper can benefit from more thorough experiments and in-depth study.

**Questions:**

I expect the experimental results on other scenarios.

---

> ### Author Response · Authors · 2024-11-21
>
> Thank you for your valuable and constructive feedback. We would like to respond to your concerns individually as follows:
>
> 1. We recognize the limitations of our work. Resource constraints and the lack of suitable existing datasets have hindered our ability to scale up at this time. Nonetheless, we have taken the crucial first step of proposing the task and validating the feasibility of our approach. We believe that multi-action video generation will and needs to be paid more attention in the future. While pioneering this direction and proposing a perfect solution is impracticable, we hope such a bi-directional unsupervised training pipeline can be valued and spur further expansion on this topic.
> 2. For the baselines mentioned, we appreciate you pointing out these relevant works. We didn't include them because our work was actually done before their release. We would like to include these baselines in the next updated version of our paper.
> Here we show a couple of quantitative results of these two models' ability in connecting action transitions. The videos are updated in the supplementary material.
>
>     As can be seen from the result, they face the same fetal issues as Dynami-Crafter and SEINE have -- the generated video is rather a smooth scene transition than a natural action transition that obeys the physical law. This is because these methods only consider image conditioning, not video conditioning, thereby neglecting temporal coherence and naturalness. As far as we know, we are the first to implement the video-based transition generation.  And we do hope our exploration and introductory verification can draw more attention to this topic.

---

> ### Comment · Reviewer_4GmR · 2024-11-27
>
> I appreciate the authors' detailed replies that clarifies much confusion. I raise my score. I agree that video transition (scene transition) is another important and unexplored task in video completion. I also agree the novelty of this work.

---

### Meta-Review · Area_Chair_8oo6 · 2024-12-21

**Metareview:**

Despite the interesting problem context and the proposed MAVIN model, the reviewers' concerns about limited novelty, insufficient experimental validation and lack of clarity. Therefore the paper's contributions are considered insufficient for acceptance on ICLR, and a rejection decision is recommended. The reviewers encourage the authors to consider expanding their experimental evaluation, providing clearer explanations and analyses, and exploring more innovative approaches to further strengthen their work.

**Additional Comments On Reviewer Discussion:**

Most reviewers engaged in discussions with the authors and part of concerns are addressed by authors except that Reviewer UVGm did not reply to the authors' response. However, novelty and sufficiency of experiments are still considered insufficient for acceptance on ICLR.

---

### Decision · Program_Chairs · 2025-01-22

Reject